



# Collective risk modelling for understanding the correlation between multi-peril accumulated losses

Toby P. Jones[1], David B. Stephenson[1], and Matthew D. K. Priestley[1]

[1]Department of Mathematics & Statistics, University of Exeter, Exeter, United Kingdom

**Correspondence:** Toby P. Jones (tpj201@exeter.ac.uk)

**Abstract.** Hazards such as storms can create multiple perils, such as windstorms and floods, that have correlated annual losses. To better understand the drivers of such correlations, this study explores three collective risk frameworks with varying complexity.

5 Mathematical expressions are derived explaining how this correlation depends on parameters such as event dispersion (clustering), and the joint distribution of the two hazard variables. Hazard variables are first assumed independent, inducing a positive correlation due to the shared positive dependence on the total number of events. The next framework allows for correlation between the hazard variables, which can then capture negative correlation between accumulated losses. The final framework builds on this by allowing for between-year correlation caused by interannual modulation of the hazard variables.

These frameworks are illustrated using European windstorm gust speeds and precipitation reanalyses from 1980-2000. They are used to diagnose why the correlation between annual wind and precipitation severity indices decreases as thresholds are increased. Only the framework with interannual modulation of the hazard variables quantitatively captures the negative correlations over Europe at high threshold. We propose that one plausible driver for the modulation is the transit time that storms

15 spend near locations.

## 1 Introduction

Environmental hazards can often lead to co-occurring perils. For example, extratropical cyclones can lead to losses from co-occurring extreme wind gusts and floods (Raveh-Rubin and Wernli, 2015; Martius et al., 2016; Owen et al., 2021) as well

20 as from storm surges (Kendon and McCarthy, 2015). Such events are also referred to as *multivariate* events since the losses result from extremes in multiple hazard variables (Zscheischler et al., 2020). Other examples include high temperatures and low precipitation leading to wildfire in south Australia (Richardson et al., 2022); storm surge and high precipitation leading to flooding after hurricanes (Juárez et al., 2022) and the combined effect of a co-occurring heatwave and drought in Africa and Asia (Wang et al., 2023). The impact from these multi-peril events is often greater than from the sum of impacts from the

25 hazards separately (Hillier and Dixon, 2020).





Multivariate compound weather hazards are receiving increasing attention in studies using a variety of statistical methods. Examples include copulas (Manning et al., 2024); comparing co-occurrence relative to a bootstrapped event set (Hillier et al., 2025), or use of extremal dependency measures (Zscheischler et al., 2021; Owen et al., 2021). These methods generally aim
to quantify the dependence between hazard variables of individual events rather than diagnose the drivers of such dependence (Hillier et al., 2020; Bevacqua et al., 2021).

In addition to the individual risk of loss due to single events, it is important for risk managers to also understand the collective
risk due to a set of events over the time period that is insured. The annual aggregation over the calendar year from January to December is particularly relevant to the insurance industry as it aligns with typical reinsurance contract timelines (Čížek et al., 2005). Collective risk not only depends on the individual risk for each event but also on properties such as temporal clustering of the events (Mailier et al., 2006; Vitolo et al., 2009; Hunter et al., 2015). Despite increasing numbers of studies on clustering, much less research has been published on the collective risk of multivariate hazards. It is common practice in insurance to
model perils separately and then assume that annual losses from different perils are independent. For example, yearly losses from wind and flood in Europe are modelled separately and then assumed to be uncorrelated (Hadzilacos et al., 2021).

To better understand the correlation between accumulated losses from different perils, this study explores and tests various collective risk modelling frameworks for diagnosing the drivers of such correlation. The methods are demonstrated by apply-
45 ing them to annually aggregated wind and precipitation severities caused by extratropical cyclones over the North Atlantic and Europe from 1980-2020. In particular, we use the frameworks to diagnose the correlation noted between annual wind and precipitation severities that was recently presented in (Jones et al., 2024).

Damage from both extreme wind and precipitation can occur within the same season (Kendon and McCarthy, 2015). As
such the annual cost of extratropical cyclone damage in Europe often reaches billions of Euros (Cusack, 2023). Consequently, protection against wind damage constitutes over 15% of global reinsurance purchases (Mitchell-Wallace et al., 2017), while the UK needed a not-for-profit flood re-insurance scheme to keep consumer premiums affordable (Browning, 2020). As large loss events are more likely to cluster (Vitolo et al., 2009; Priestley et al., 2018; Renggli and Zimerli, 2016), understanding the relationship between annual wind and precipitation hazards from extratropical cyclones is crucial for re-insurers to best
diversify their risk across hazards (Gro, 2005; Klugman et al., 2019).

Hillier and Dixon (2020) found a positive relationship between seasonally aggregated extreme wind gusts and precipitation, the wind hazard increases during the wettest years for most of Europe. Similarly, positive correlation was found to exist between wind and precipitation aggregated across the UK from daily to seasonal timescales (Bloomfield et al., 2023). However,
Jones et al. (2024) found that the positive correlation between annual wind and precipitation severities decreased and even





became slightly negative over Europe for more extreme severities as thresholds for the hazard variables were increased.

This study aims to answer the following questions:

– What assumptions are required for a collective risk model to be able to capture the correlation between aggregate losses at all spatial locations?

– Can such a collective risk model quantitatively account for how the correlation changes for more extreme events?

– What are the key drivers of the changes in correlation with threshold?

Section 2 presents three collective risk models of increasing complexity and shows how the correlation of aggregate losses depends on parameters such as overdispersion (clustering), skewness of the hazard variables, and correlation between the individual hazard variables. Section 3 then applies and tests the frameworks on the storm data used in Jones et al. (2024). Conclusions and ideas for future work are presented in Section 4.

## 2  Collective risk modelling

### 2.1  Severity Indices

The damage or loss at a given location is often approximated to be a function of the hazard variable, i.e. $g(X)$ where $X$ is the hazard variable (e.g. wind gust speed). Idealised forms of these functions are known as Severity Indices (SI). For example, this study uses a simple *exceedance over threshold* SI defined following Jones et al. (2024) as

$$g(X) = \begin{cases} X - u_X & X_i > u_X \\ 0 & X_i \leq u_X. \end{cases} \tag{1}$$

The threshold $u_X$ can be a fixed value for all locations (e.g. $20\text{ms}^{-1}$; Jones et al. (2024)) or a percentile of $X$ that varies with location (e.g. $u_X = X_{0.98}$; Klawa and Ulbrich (2003)).

### 2.2  Aggregate Severity Indices

Accumulated losses over a given time period (e.g. a year) are then approximated by the random sum of Severity Indices $g(X_i)$ over the set of events $i = 1, 2, \ldots, N$ that occurred in the period. Therefore the Aggregate Severity Index (ASI) is defined as

$$S_X = \begin{cases} g(X_1) + g(X_2) + \ldots + g(X_N) & N > 0 \\ 0 & N = 0. \end{cases} \tag{2}$$





where $\{X_1, \ldots, X_N\}$ are the hazard variables for each of the $N$ events. The distribution of $S_X$ determines the collective risk of accumulated losses over the chosen time period.

In this study, we shall consider events that have perils caused by two hazard variables $X$ and $Y$ with thresholds $u_X$ and $u_Y$, respectively, resulting in annual ASI $S_X$ and $S_Y$. The total number of events, $N$, only includes events that increase $S_X$ or $S_Y$ (or both) i.e. events where $X > u_X$ or $Y > u_Y$. For simplicity of notation, we shall refer to $X' = g(X)$ and $Y' = g(Y)$ simply as $X$ and $Y$, respectively.

### 2.3 ASI modelling frameworks

An ASI is the sum of a random number $N$ of random variables $\{X_1, X_2, \ldots, X_N\}$ and is known as a *random sum* in actuarial science (Ambagaspitiya, 1999; Ren, 2012; Tang, 2001). Its distributional properties depend on the distribution of $N$, the distribution of the $X$ variables, and the joint distribution between $N$ and the $X$. For example, the expectation ($\mathrm{E}[S_X]$) and variance ($\mathrm{var}(S_X)$) of random sums have been derived long ago (Wald (1945);Blackwell and Girshick (1947)). These results assume that the $X$ variables are independent of $N$, but this assumption can be relaxed (Cohen, 2019).

Far less attention has been given to correlations between random sums. Mirzai (1999) used a simplified approach relying on counts of extremes, although counts were restricted to follow a Poisson distribution (something uncharacteristic of European windstorms, Mailier et al. (2006)). Kolev and Paiva (2008) defined a more flexible framework, but this includes $S_X$ and $S_Y$ as explanatory variables. Neither study applied their respective frameworks to real-world hazards. Only Jones (2022) has applied a similar correlation framework to model hazard data, but this focused on frequency and aggregate wind hazard.

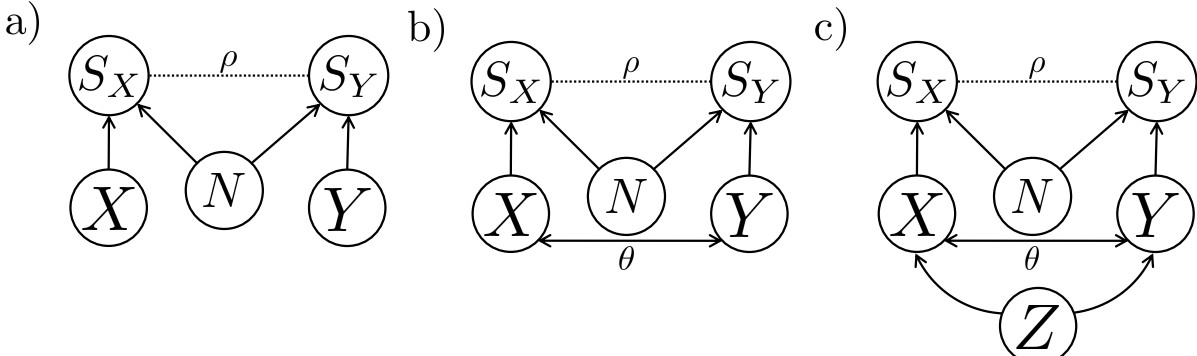

**Figure 1.** Dependence assumptions for the three frameworks. Arrows show which variables can causally influence others.

This study considers 3 frameworks of increasing complexity. For simplicity, all the frameworks assume that severities are independent of the counts, i.e., $\mathrm{Cov}(N, X_i) = 0$ and $\mathrm{Cov}(N, Y_i) = 0$ for $i \in \{1, 2, \ldots, N\}$. The three frameworks are:





– **Framework A: Uncorrelated hazard variables**.

Assumes $\text{Cov}(X_i, Y_j) = 0$, $\text{Cov}(X_i, X_j) = \sigma_X^2 \delta_{ij}$ and $\text{Cov}(Y_i, Y_j) = \sigma_Y^2 \delta_{ij}$ for all $i,j \in \{1, 2 \ldots, N\}$ where $\delta_{ij} = 1$ if $i = j$ and 0 otherwise. The standard deviations for hazards $X$ and $Y$ are $\sigma_X$ and $\sigma_Y$. These are common assumptions often used by actuaries (Kaas et al., 2008).

– **Framework B: Correlated hazard variables**.

Assumes $\text{Cov}(X_i, Y_j) = \theta \sigma_X \sigma_Y \delta_{ij}$, $\text{Cov}(X_i, X_j) = \sigma_X^2 \delta_{ij}$ and $\text{Cov}(Y_i, Y_j) = \sigma_Y^2 \delta_{ij}$ for all $i,j \in \{1, 2 \ldots, N\}$. Correlation $\theta = \text{Cor}(X_i, Y_i) \in [-1, 1]$ is the correlation between the hazard variables for each event.

– **Framework C: Correlated hazard variables modulated by $Z$**.

Assumes $\text{Cov}(X_i, Y_j) = \theta \sigma_X \sigma_Y \delta_{ij} + \text{Cov}(\overline{X}, \overline{Y})$, $\text{Cov}(X_i, X_j) = \sigma_X^2 \delta_{ij} + \text{Var}(\overline{X})$ and $\text{Cov}(Y_i, Y_j) = \sigma_Y^2 \delta_{ij} + \text{Var}(\overline{Y})$ for all $i,j \in \{1, 2 \ldots, N\}$ where $\overline{X} = E(X|Z)$ and $\overline{Y} = E(Y|Z)$. Variable $Z$ is a latent variable that is considered to vary between but not within years.

The dependency structure of each framework is summarised in Figure 1. Framework B is a special case of framework C having $\text{var}(Z) = 0$ (i.e. no interannual modulation), and framework A is a special case of framework B having $\theta = 0$ (i.e. no hazard correlation).

Using these assumptions it is possible to derive the correlation $\rho = \text{Cor}(S_X, S_Y)$ between the aggregate severities for each of the three frameworks (see Appendix). For framework A, one obtains

$$\rho_A = \frac{\phi J_X J_Y}{\sqrt{(1 + \phi J_X^2)(1 + \phi J_Y^2)}}. \tag{3}$$

where $\phi = \text{var}(N)/E[N]$ is the dispersion in counts (Mailier et al., 2006), and $J_X = E[X]/\sqrt{\text{var}(X)}$ and $J_Y = E[Y]/\sqrt{\text{var}(Y)}$ are the signal-to-noise ratios of the two hazard variables. The correlation of framework A is always non-negative and increases from 0 to 1 as the dispersion $\phi$ goes from 0 to $\infty$, and hence large amounts of clustering $\phi \gg 1$ induce high correlation between the aggregated severities of the two perils. Framework B gives

$$\rho_B = \frac{\theta + \phi J_X J_Y}{\sqrt{(1 + \phi J_X^2)(1 + \phi J_Y^2)}}. \tag{4}$$

where $\theta = \text{Cor}(X_i, Y_i)$ is the correlation between hazard variables for individual events. Unlike framework A, framework B can produce negative correlations provided $\theta < \phi J_X J_Y$. It should be noted that $\rho \geq \theta$ when $\theta < 0$ and so $\rho$ can never be more negative than $\theta$. Framework C gives

$$\rho_C = \frac{\theta + \phi J_{XY} + \lambda K_{XY}}{\sqrt{(1 + \phi J_X^2 + \lambda K_X^2)(1 + \phi J_Y^2 + \lambda K_Y^2)}} \tag{5}$$

$$= \frac{\theta}{\sqrt{d}} + \frac{\phi J_{XY}}{\sqrt{d}} + \frac{\lambda K_{XY}}{\sqrt{d}}$$



where $\lambda = \mathrm{E}[N]$, $d = \sqrt{(1 + \phi J_X^2 + \lambda K_X^2)(1 + \phi J_Y^2 + \lambda K_Y^2)}$ and

$$J_{XY} = \frac{E_Z\big[\mathrm{E}[X|Z]\mathrm{E}[Y|Z]\big]}{\sqrt{E_Z\big[\mathrm{var}(X|Z)\mathrm{var}(Y|Z)\big]}} \qquad\qquad K_{XY} = \frac{\mathrm{Cov}\big(\mathrm{E}[X|Z], \mathrm{E}[Y|Z]\big)}{\sqrt{E_Z\big[\mathrm{var}(X|Z)\mathrm{var}(Y|Z)\big]}}$$

$$J_X^2 = \frac{E_Z\big[\mathrm{E}[X|Z]^2\big]}{E_Z\big[\mathrm{var}(X|Z)\big]} \qquad\qquad J_Y^2 = \frac{E_Z\big[\mathrm{E}[Y|Z]^2\big]}{E_Z\big[\mathrm{var}(Y|Z)\big]}$$

$$K_X^2 = \frac{\mathrm{var}\big(\mathrm{E}[X|Z]\big)}{E_Z\big[\mathrm{var}(X|Z)\big]} \qquad\qquad K_Y^2 = \frac{\mathrm{var}\big(\mathrm{E}[Y|Z]\big)}{E_Z\big[\mathrm{var}(Y|Z)\big]}.$$

The variable $Z$ is a latent variable that is considered to vary with the years and so then $\mathrm{E}[X|Z]$ are the annual means of $X$, and $\mathrm{Cov}\big(\mathrm{E}[X|Z], \mathrm{E}[Y|Z]\big)$ is the covariance between the annual means of $X$ and $Y$. This framework has the advantage that the interannual modulation can allow $\rho$ to be more negative than $\theta$.

## 3 Data example: correlation of wind and precipitation storm severity indices

### 3.1 Storm data 1980-2020

This frameworks in this study are applied to the same data and cyclone extraction procedures as detailed in Jones et al. (2024). The cyclones and hazard variables are extracted from 1-hourly ERA5 reanalysis from 1980-2020 (Hersbach et al., 2020). This hourly data, with $0.25°$ spatial resolution, captures many of the smaller scale features that contribute to extreme wind and precipitation (Whitford et al., 2023; Browning, 2004; Kodama et al., 2019).

Cyclones are first identified and tracked at 850hPa using the TRACK algorithm (Hodges, 1994). This is a widely adopted method for cyclone tracking (Hawcroft et al., 2012; Manning et al., 2023; Priestley and Catto, 2022; Maddison et al., 2020; Yu et al., 2023; Hay et al., 2023) and performs similarly to other tracking algorithms (Bourdin et al., 2022). A constant $5°$ radius is applied around the tracks to determine the influence of the cyclone, which is comparable to what has been used in previous studies (Hawcroft et al., 2015; Kodama et al., 2019).

Severity indices were created using Eqn. 1 at each grid point for each cyclone using the 3-second maximum wind gust speeds ($x_i$) and total accumulated precipitation ($y_i$) over the duration that each cyclone was within $5°$ of that grid point. Annual ASIs $S_X$ and $S_Y$ were then created for every grid point and every calendar year (1 January-31 December) using Eqn. 2.

### 3.2 Framework skill at modelling correlation

Following Jones et al. (2024), Figures 2a)-c) show sample correlation values at each grid point between wind and precipitation ASIs for different threshold levels. Strong positive correlation occurs across almost all of domain when thresholds are zero (Fig. 2a). Correlations then reduce for higher thresholds, with regions of negative correlation appearing mostly over land (Fig. 2b). At the highest thresholds, negative correlation becomes more widespread across land and starts to appear in isolated loca-





tions in the Atlantic Ocean (Fig. 2c).

**Figure 2.** Sample correlation between $S_X$ and $S_Y$ (a-c) and estimates of the correlation from framework A (d-f), framework B (g-i), and framework C (j-l). Columns represent different threshold combinations for $(u_X, u_Y)$: no thresholds $(0\mathrm{ms}^{-1}, 0mm)$ (left), $(10\mathrm{ms}^{-1}, 10mm)$ (centre) and high thresholds $(20\mathrm{ms}^{-1}, 20mm)$ (right).

It is of interest to see how well the frameworks capture these correlations at different thresholds. Correlations calculated using these frameworks are shown in the panels below: Fig. 2d)-f) (framework A), Fig. 2g)-i) (framework B), and Fig. 2j)-l) (framework C). All the frameworks perform similarly well at capturing the positive correlation when the thresholds are zero (Figs. 2d, 2g, and 2j). Framework A shows a decrease in correlation at higher thresholds (Figs. 2e and 2f), but is unable to

170 produce any of the negative correlations seen in the sample correlations (Figs. 2b and 2c). Framework B shows greater decrease at higher threshold (Figs. 2h and 2i) with some small negative correlations appearing but still not as negative as in the sample correlations. Framework C shows a stronger decrease (Figs. 2k and 2l) with much more extensive negative correlations over




land at the highest threshold. The location of negative correlation over the northwest of mainland Europe is correctly captured at the highest thresholds (Figure 2k). In summary, Framework C is the only framework able to capture the correlations at each of the thresholds.

### 3.3 Analysis of components in Framework C

The correlation in Equation (5) is the sum of 3 components each having the same denominator $d$. The terms can be interpreted as follows:

- Within-year dependency: $\theta$, the simultaneous correlation between wind and precipitation in each event.

- Event dispersion: $\phi J_{XY}$, the positive dependence induced in $S_X$ and $S_Y$ by their positive relationships with counts. Larger dispersion ($\phi$) in annual counts leads to a greater effect.

- Interannual dependency: $\lambda K_{XY}$, the relationship between yearly mean values of wind and precipitation, scaled by the mean number of events per year ($\lambda$).

Figure 3 shows the decomposition of the correlation $\rho$ for framework C. The event dispersion component is positive at the lowest threshold but decreases towards zero for higher thresholds (Figs. 3g-i). It is the main contributor to $\rho$ at low thresholds as can be seen in the similarity between Fig. 3g and 3a. The within-year dependency component is also positive at the lowest threshold (3g), but decreases to negative over Europe at the highest threshold (3i). The interannual dependency component follows a similar pattern but with a stronger decrease to more negative values at high values of threshold (3j-l). The interannual dependency component is the main contributor to $\rho$ at high thresholds as can be seen in the similarity between Fig. 3l and 3c.

For more detail on how each of the components varies with threshold, Figure 4 shows the components versus threshold for a region covering France (red box Fig. 5b). The threshold is set to the same value for both wind and precipitation. The SI for the region is calculated by summing the SI over all land and sea grid points in [4.75°W-8.5°E, 42.25°-51.75°N]. The number of storms is calculated for the entire region by counting the number of events when SI is positive at one or more of the grid points. Correlation, $\rho$, decreases with increasing threshold and goes negative above $18\,\mathrm{ms}^{-1}$ and 18mm (red line Figure 4). This behaviour is generally well captured by the framework (blue dashed line). The positive within-year dependency component (solid thin line) is largely compensated at all thresholds by the negative within-year dependency component (thin dashed line). This results in the framework correlation closely following the interannual dependency component (thin dotted line).

### 3.4 A potential driver: storm transit duration

Framework C introduced latent variable $Z$ that was considered to be an interannual modulator of wind severity $X$ and precipitation severity $Y$. It is of interest to speculate as to what this driver $Z$ might be. One obvious candidate is how fast storms





**Figure 3.** Decomposition of the correlation for framework C: the framework correlation $\rho$ (a-c), $\phi J_{XY}/\sqrt{d}$ (d-f), $\theta/\sqrt{d}$ (g-i), $\lambda K_{XY}\sqrt{d}$ (j-i). Columns represent different threshold combinations for $(u_X, u_Y)$: no thresholds $(0\text{ms}^{-1}, 0mm)$ (left), $(10\text{ms}^{-1}, 10mm)$ (centre) and high thresholds $(20\text{ms}^{-1}, 20mm)$ (right)





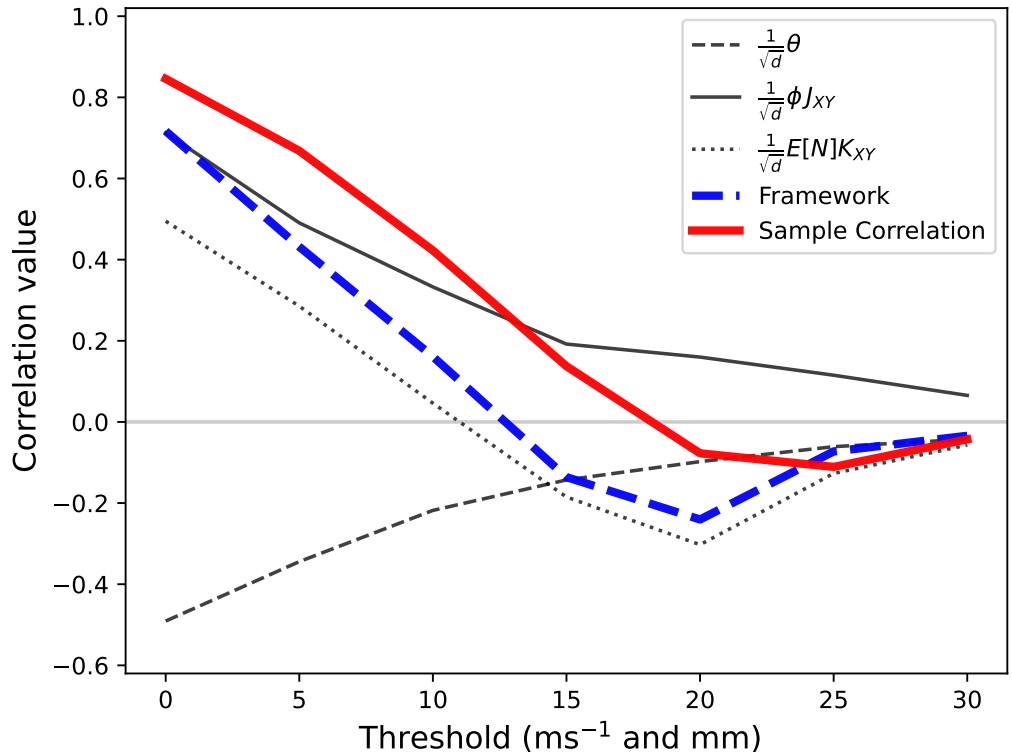

**Figure 4.** Framework C correlation and its components for France (red box in Figure 5) over a range of threshold levels where $u_X = u_Y$. Thick lines represent sample correlation (red solid line) and framework estimate (blue dashed line). Thinner lines are each component of the framework: $\theta/\sqrt{d}$ (thin solid line),$\phi J_{XY}/\sqrt{d}$ (thin dashed line), $\lambda K_{XY}/\sqrt{d}$ (thin dotted line).

propagate past each grid point location (Hillier and Dixon, 2020; Rhodes, 2017). For a constant precipitation rate, slower mov-
205 ing systems will have more time to precipitate at a fixed location and so will lead to larger precipitation totals. Slow moving
windstorms and a weaker jet stream are conducive to precipitation-only extremes (Manning et al., 2024). One might also expect
slower storms to be ones that do not have the strongest local wind speeds. Indeed, such behaviour can be seen, for example, in
the values of $X_i$ and $Y_i$ shown for France in Fig. 4a. Grid point events with total rainfall exceeding 100mm have long durations
exceeding 40 hours, whereas events with extreme wind speeds exceeding $37\mathrm{ms}^{-1}$ have much shorter durations, typically less
than 20 hours. Storm duration here is defined as the number of hours a storm track is within $5°$ of the individual grid point.
Furthermore, the longest durations (slowest propagation speeds) occur at lower intermediate wind speeds of $5 - 30\mathrm{ms}^{-1}$ in
agreement with Hillier and Dixon (2020). In addition to duration, the previous path of the storm affects moisture availability.
Greater poleward propagation speed can increase precipitation rates (Sinclair and Dacre, 2019). Figures 5b, c show tracks of
the 42 storms that led to extreme wind speeds $> 37\mathrm{ms}^{-1}$, and the tracks of the 57 storms that led to total precipitation $> 100$



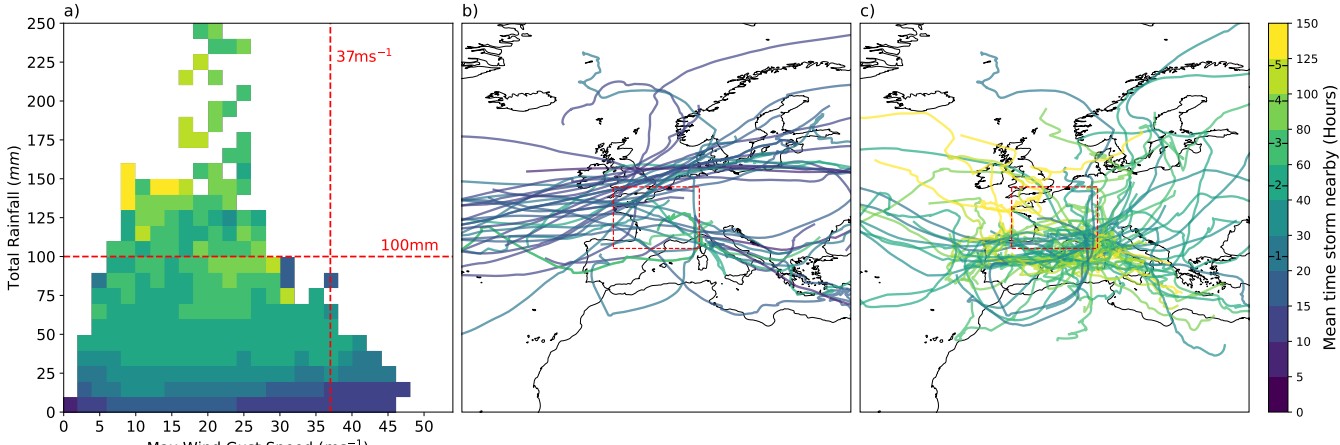

**Figure 5.** (a) Heatscatter plot of wind gust and precipitation values for France region. Boxes are coloured by mean storm duration. Red dashed lines depict thresholds of $37\mathrm{ms}^{-1}$ and $100mm$ respectively, (b) Tracks for storms where a grid point had wind speed values above $37\mathrm{ms}^{-1}$, c) Tracks for storms where a grid point had precipitation values above $100mm$. Tracks are coloured by mean storm duration with the same scale as panel a.

mm in France. The extreme wind speed storms tend to have more zonal tracks coming across the Atlantic, whereas the extreme precipitation storms are more meridional with many coming from the south over the Mediterranean. Hence, the duration of storms leading to extremes is also related to where the storms originate - longer duration ones appear to originate more from the south where there is potentially more moisture availability over the warm Mediterranean Sea.

By considering annual means of storm duration for each grid point, it is possible to investigate whether duration might be able to account for the interannual dependency between wind and precipitation. Figure 6 shows the correlations between annual mean storm duration $\overline{D} = S_D/N$ and annual mean intensities, $\overline{X} = S_X/N$ and $\overline{Y} = S_Y/N$, at different thresholds. Annual mean duration has a mostly negative correlation with annual mean wind intensity especially over European land regions (Figs. 6a-c), whereas it has a spatially more uniform positive correlation with precipitation intensity (Figure 6d-f). The magnitudes of these correlations increase over Europe for increasing threshold, which helps to account for why a negative correlation intensifies in the correlation between annual mean wind and precipitation intensities (Fig. 6g-i).

## 4 Conclusions

This study has explored collective risk frameworks to model correlation between aggregate severities that occur from multivariate compound events. It has been found that to reproduce the correlation in the wind and precipitation ASIs, it is necessary to include simultaneous correlation between the hazard variables and interannual modulation of the mean hazard variables. Of



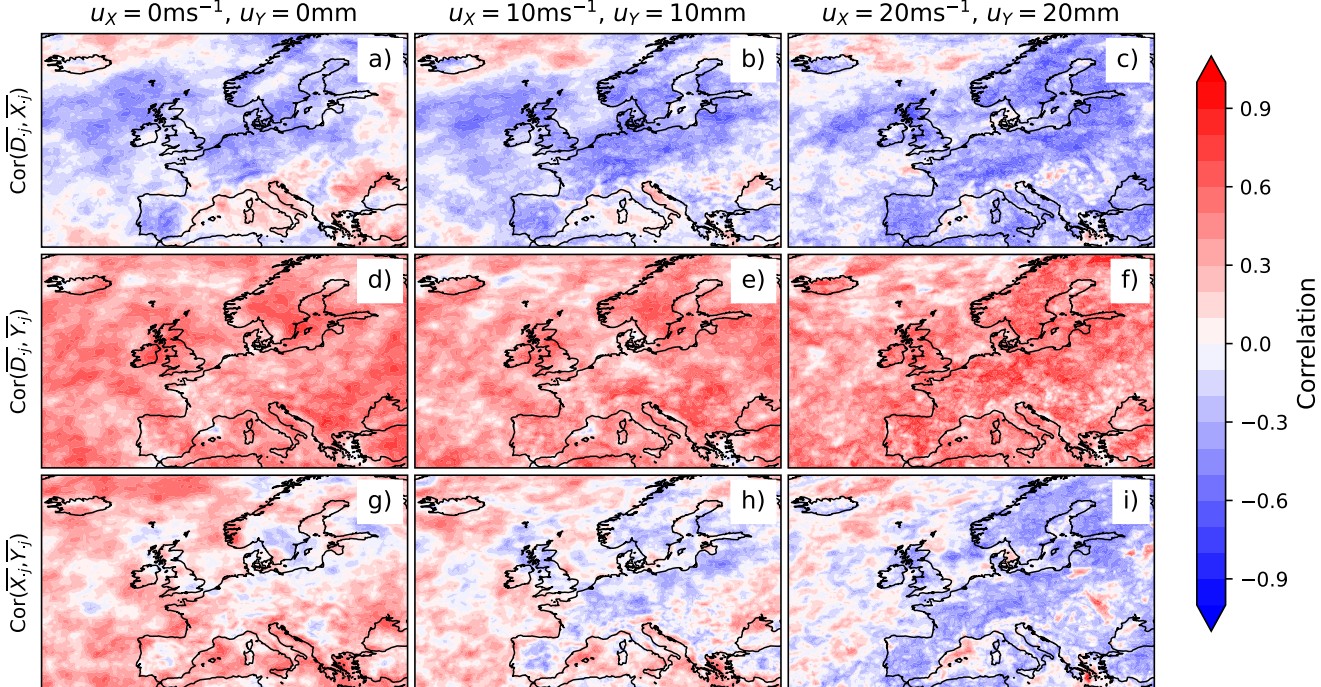

**Figure 6.** Sample correlations between mean yearly duration and mean yearly wind speed (a-c), mean yearly duration and mean yearly precipitation (d-f), mean yearly wind speed and mean yearly precipitation (g-i). Columns represent different threshold combinations for $(u_X, u_Y)$: no thresholds ($0\mathrm{ms}^{-1}, 0mm$) (left), ($10\mathrm{ms}^{-1}, 10mm$) (centre) and high thresholds ($20\mathrm{ms}^{-1}, 20mm$) (right)

the three introduced frameworks, only framework C was able to quantitatively capture the correlations across Europe and the North Atlantic at different severity thresholds, including the higher thresholds where negative correlations emerge. Framework C (and the other frameworks) assumed that the hazard variables are independent of the counts and so it does not appear necessary to include severity dependence on frequency as was considered in Hunter et al. (2015); Cohen (2019). We hypothesise that one of the possible drivers for the interannual modulation is the transit time spent by a storm near to a grid point: total precipitation increases for slower transits, whereas gust speeds tend to increase.

This study has several caveats such as:

- this study has, for simplicity, only considered correlation of ASI that are co-located at the same grid point, whereas the hazards can be displaced from one another but still lead to co-occuring losses for an insured region. Local exposure does not always result in local damage, for example, heavy precipitation at one location may cause flooding much further downstream (Viglione and Rogger, 2015);





- – the SIs used here are highly idealised loss functions - a strict cut-off is an unrealistic representation of vulnerability and therefore damage (Kaas et al., 2008);

- – absolute thresholds have been used across the whole domain. However, similar results are obtained when using relative thresholds defined by the local quantiles of the hazard variables (not shown);

- – the precipitation SI is a proxy for flood but does not contain any information about soil moisture (De Luca et al., 2017) or hydrology that are also required for flood prediction;

- – this study has chosen to use the annual aggregation period typical of insurance contracts. Use of other periods, such as individual winters, gives broadly similar results (not shown);

- – the data only spans a relatively short period of 40 years. However, examination of reanalyses going back to 1940 show broadly similar behaviour (not shown due to data quality being poorer in the pre-satellite period);

- – ERA5 precipitation is estimated rather than being observed (Hersbach et al., 2020) meaning inaccuracies can exist (Rhodes et al., 2014). Despite reasonable representation of extratropical precipitation (Lavers et al., 2022), ERA5 is not as accurate as station measurements;

- – alternative storm tracking algorithms could have been used (see Bourdin et al. (2022)) as well as other methods of defining footprints e.g. Vitolo et al. (2009) and Lockwood et al. (2022).

This research could be extended in several ways. It would be of interest to test the effect of relaxing some of the caveats such as the co-located hazard assumption. The framework could also be extended to more than two hazards, which would allow it to be used to investigate compound wind/flood/storm surge losses. Finally, the framework could be applied to output from climate change simulations to understand better how correlation between losses might change in the future. It would also be of interest to better understand what climatic conditions affect storm transit duration in different regions. The speed of the westerly jet and the North Atlantic Oscillation are likely to play key roles but there may be other factors of interest.

*Data availability.* The data that support the findings of this study are openly available in Copernicus Climate Change Service Climate Data Store at https://doi.org/10.24381/cds.bd0915c6

## Appendix A:  Appendix: Correlation between aggregated losses

Since frameworks A and B are special cases of framework C, it suffices to derive the correlation for framework C. Using the Law of Total Covariance and the independence of the hazard variables on counts $N$ allows the covariance to be decomposed



as follows:

$$
\begin{aligned}
\mathrm{Cov}(S_X, S_Y) &= \mathrm{E}_Z\big[\mathrm{Cov}(S_X|Z, S_Y|Z)\big] + \mathrm{Cov}_Z(\mathrm{E}[S_X|Z], \mathrm{E}[S_Y|Z]) \\
&= \mathrm{E}_Z\big[\mathrm{E}[N]\mathrm{Cov}(X_i|Z, Y_i|Z) + \mathrm{Var}(N)\mathrm{E}[X_i|Z]\mathrm{E}[Y_j|Z]\big] + \mathrm{Cov}_Z(\mathrm{E}[N]\mathrm{E}[X_i|Z], \mathrm{E}[N]\mathrm{E}[Y_j|Z]) \\
&= \mathrm{E}[N]\mathrm{E}_Z\big[\mathrm{Cov}(X_i|Z, Y_i|Z)\big] + \mathrm{Var}(N)\mathrm{E}_Z\big[\mathrm{E}[X_i|Z]\mathrm{E}[Y_j|Z]\big] + \mathrm{E}[N]^2\mathrm{Cov}_Z(\mathrm{E}[X_i|Z], \mathrm{E}[Y_j|Z]).
\end{aligned}
$$

Using the Law of Total Variance, the variance can be decomposed as

$$
\begin{aligned}
\mathrm{Var}(S_X) &= \mathrm{E}_Z\big[\mathrm{Var}(S_X|Z)\big] + \mathrm{Var}_Z(\mathrm{E}[S_X|Z]) \\
&= \mathrm{E}_Z\big[\mathrm{E}[N]\mathrm{Var}(X_i|Z) + \mathrm{Var}(N)\mathrm{E}[X_i|Z]^2\big] + \mathrm{Var}_Z(\mathrm{E}[N]\mathrm{E}[X_i|Z]) \\
&= \mathrm{E}[N]\mathrm{E}_Z\big[\mathrm{Var}(X_i|Z)\big] + \mathrm{Var}(N)\mathrm{E}_Z\big[\mathrm{E}[X_i|Z]^2\big] + \mathrm{E}[N]^2\mathrm{Var}_Z(\mathrm{E}[X_i|Z]) \\
&= \mathrm{E}[N]\big(\mathrm{E}_Z\big[\mathrm{Var}(X_i|Z)\big] + \phi\mathrm{E}_Z\big[\mathrm{E}[X_i|Z]^2\big] + \mathrm{E}[N]\mathrm{Var}_Z(\mathrm{E}[X_i|Z])\big).
\end{aligned}
$$

and a similar expression is obtained for $\mathrm{Var}(S_Y)$. Therefore the correlation between $S_X$ and $S_Y$ can be written as

$$
\begin{aligned}
\rho &:= \frac{\mathrm{Cov}(S_X, S_Y)}{\sqrt{\mathrm{Var}(S_X)\mathrm{Var}(S_Y)}} \quad \text{(Definition of correlation)} \\
&= \frac{\theta + \phi J_{XY} + \lambda K_{XY}}{\sqrt{(1 + \phi J_{XX} + \lambda K_{XX})(1 + \phi J_{YY} + \lambda K_{YY})}}
\end{aligned}
\tag{A1}
$$

where

$$
\begin{aligned}
\lambda &= \mathrm{E}[N] \\
\phi &= \frac{\mathrm{Var}(N)}{\mathrm{E}[N]} \\
\theta &= \frac{\mathrm{E}_Z\big[\mathrm{Cov}(X_i|Z, Y_i|Z)\big]}{\sqrt{\mathrm{E}_Z\big[\mathrm{Var}(X_i|Z)\big]\mathrm{E}_Z\big[\mathrm{Var}(Y_i|Z)\big]}} \\
J_{XY} &= \frac{\mathrm{E}_Z\big[\mathrm{E}[X_i|Z]\mathrm{E}[Y_i|Z]\big]}{\sqrt{\mathrm{E}_Z\big[\mathrm{Var}(X_i|Z)\big]\mathrm{E}_Z\big[\mathrm{Var}(Y_i|Z)\big]}} \\
K_{XY} &= \frac{\mathrm{Cov}_Z(\mathrm{E}[X_i|Z], \mathrm{E}[Y_j|Z])}{\sqrt{\mathrm{E}_Z\big[\mathrm{Var}(X_i|Z)\big]\mathrm{E}_Z\big[\mathrm{Var}(Y_i|Z)\big]}}.
\end{aligned}
$$

The correlation for framework B is obtained by setting all the $K$ terms to zero and $J_{XY} = J_X J_Y$, $J_{XX} = J_X^2$, $J_{YY} = J_Y^2$ (because $\mathrm{E}[X_i|Z] = \mathrm{E}[X_i]$ and $\mathrm{E}[Y_i|Z] = \mathrm{E}[Y_i]$ are constants and so no longer vary or co-vary). Framework A correlation is obtained from that of framework B by simply setting the event correlation $\theta$ to zero.





The parameters in the model are estimated by replacing expectations by sample means:

$$\lambda = \mathrm{E}[N] \to \frac{1}{T}\sum_{t=1}^{T} N_t = \overline{N}$$

$$\mathrm{Var}(N) \to \overline{N^2} - (\overline{N})^2$$

$$\mathrm{E}_Z\big[\mathrm{E}[X_i|Z]\mathrm{E}[Y_i|Z]\big] \to \overline{(S_X/N)(S_Y/N)}$$

$$\mathrm{Cov}_Z(\mathrm{E}[X_i|Z],\mathrm{E}[Y_j|Z]) \to \overline{(S_X/N)(S_Y/N)} - (\overline{S_X/N})(\overline{S_Y/N})$$

$$\mathrm{E}_Z\big[\mathrm{Cov}(X_i|Z,Y_i|Z)\big] \to \overline{S_{XY}/N} - \overline{(S_X/N)(S_Y/N)}$$

where $t = 1, 2, \ldots, T$ is the year and $N$, $S_X$, and $S_Y$ are the counts and ASI and $S_{XY} = \sum_{i=1}^{N} X_i Y_i$ for each of the years. Sample means of the quantities involving sums divided by $N$ were only taken over the years when $N_t > 0$.

$\mathrm{ms}^{-1}$

*Author contributions.* TJ and DS devised the methodology and investigated different frameworks. MP created the storm footprint dataset. TJ conducted the analysis of frameworks at different thresholds, produced all figures and wrote original draft. Supervision and guidance of this was provided by DS and MP. All authors reviewed and edited the manuscript.

*Competing interests.* The authors declare no competing interests.

*Acknowledgements.* T. P. Jones thanks for funding from the Engineering and Physical Sciences Research Council [grant no: EP/R513210/1].
310    M. D. K. Priestley thanks funding from the WTW Research Network.



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
