# Peer review of "Collective risk modelling for understanding the correlation between multi-peril accumulated losses"

_EGUsphere, 2025_

## Referee Comment (RC2)

**Collective risk modelling for understanding the correlation between multi-peril accumulated losses**
Toby Jones, David Stephenson, and Matthew Priestley

This paper presents and explores three proposed risk frameworks for quantifying co-occurring risk, particularly with a focus on European windstorms. An interesting and necessary investigation into how correlation is measured and interpreted. A motivation for this is a discrepancy between the findings of past work, generally indicating a positive correlation between flooding and wind, and one of the authors' findings of a negative correlation for seasonally aggregated losses. Whilst the work appears sound, I recommend moderate revisions to increases the accessibility and communicative power of the work.

I have two specific suggestions, and a range of other comments below which are primarily aimed at increasing the transparency of the work and ability to link it to related papers.

1. Please concisely expand the context provided to give appropriate credit to past work, which is thin in some places (e.g. Sections 2.1 & 2.2).
2. To increase the readability of the work and make it accessible to a wider audience, I think that an additional Figure panel is needed to illustrate the data underlying the negative correlation i.e. a scatter plot showing the ASI data points for a particular (x,y) location, the two thresholds from Fig2 a-c, different colours for dots contributing to Fig. 2a-c at the location. Consider adding a trend line for the correlation(s). And, also mark this location on Fig2 a-c. This might be a second panel in Fig. 4.

A large caveat to this review is that my undergraduate training is neither as a mathematician nor statistician, and as such that I cannot guarantee an in-depth critique of the technical part of this paper (i.e. the derivation in Appendix A).

Comments on the manuscript

**Abstract**

L7 – Here, please make it clear that the assumption is of the same number of events for each hazard. Why might this be appropriate for extreme wind, precipitation and flooding - particularly when they exceed a threshold? This can be discussed later. The number of impactful flooding and wind events expected in the UK are, for instance, expected to differ (e.g. see Hillier et al 2024 - doi:10.5194/gc-7-195-2024). If I have mis-understood, please use this opportunity to make the paper more widely accessible i.e. is this handled by the distribution shapes of X and Y.

L14 – Please fix the grammar here. Perhaps, '… high thresholds'.

L57 – Is it important that the metrics used for 'correlation' in this paragraph are mentioned, as the difference between how these measures change in the same (or different ways) is an important to the interpretation of the models in this paper in the context of previous work. To illustrate that metric matters, it is perfectly possible for spearman's correlation to decrease as tail impact increases (i.e. see material at https://doi.org/10.5281/zenodo.12533412, Q11&12 on Task 2). Hillier et al (2025) – 10.1002/JOC.8763 – have also quantified co-occurrence and use Uplift.

L57 – Another detail that needs to be noted to allow the reader to compare this work to previous work is how annual/seasonal are defined i.e. Oct-Mar or similar in previous work, and Jan-Dec here, effectively taking part of two storm seasons.

L70 – Applying a correlation between hazard variables in order to understand potential losses is not, as far as I am aware common. It is done in one context by Hillier (2025). Please add references to put this choice of approach in context, particularly if they've been missed by Hillier (2025).

**Models**

L77 – The phrase '… following Jones et al (2024) ….' implies that Jones is the originator of this approach of SI over a threshold.  Please add references and rephrase to better reflect the origin of the approach.

L79 – Similarly, the 20 ms$^{-1}$ threshold is widely used and sensitivity tested (e.g. in Hillier & Dixon, 2020) although it long predates this.  Please add references and rephrase to better reflect the origin of the approach.

L83 – Similarly, for Section 2.2 concisely add some background for the use of aggregate SIs for wind and cyclones.

L105 – Fig. 1 is very helpful to improve accessibility. Thank you.

L115 – Please clarify in words whether Z modulates X and Y in the same way (i.e. up or down), or whether it is possible for Z to affect X and Y differently.

**Application**

L160 – Following, or replicated from? Please clarify to make it clear whether Figure 2 is new work.

L163 – Please add a sentence here on whether or not the spatial pattern of high and low correlations match previous work (e.g. Martius, 2016; Hillier & Dixon, 2020).  If there is a match, it will then be clear to the reader that there is agreement spatially, with the primary debate about the magnitude & sign of the correlation.

L193 – Red box for France.  Is this 5° from the central point?  If not, it would be better for comparison and consistency across the manuscript to make this match the approach on L157, and clarify there that it's a box not a radius.

L206 – This jet stream connection was first published in Hillier & Dixon (2020) - Dixon's idea. Please consider add the reference to this point.

L235 – Fix brackets for references.

L251 – To link this work into that in the introduction, it is probably worth showing the Oct-Mar results in Supplementary Material so that readers can decide how to link this to earlier work.

**Figures**

Figure 2 – On a-c, please add stippling or similar to indicate areas where the observations have statistical significance (i.e. can be safely interpreted as existing).  And, in the caption, decide if units are in italics or not.

Figure 4 – Add error bars, or similar, to the sample correlation curve please.

Figure 5 – The two sets of numbers on the scale bar reduce clarity.  Please find another solution.  However, I like this figure; interesting similarities to Fig. 2g of Hillier & Dixon (2020) showing different wind directions for 'windy' and 'wet'.

Figure 6 – Please add stippling or similar to indicate where the correlations are statistically significant (or not).

---

## Author Comment (AC1)

Response to Anonymous Referee #2

We would like to thank the reviewer for their helpful comments on our manuscript. Below we list how we've addressed each of their comments (in blue italics).

*Please concisely expand the context provided to give appropriate credit to past work, which is thin in some places (e.g. Sections 2.1 & 2.2).*

We have expanded these sections with more detail. Section 2.1 has had L80-L85 added:

*Numerous SIs have been created for wind damage. Klawa and Ulbrich (2003) use the cube of wind gust above the local 98th percentile, with numerous other studies (e.g. Leckebusch et al. (2007, 2008); Pinto et al. (2012); Little et al. (2023)) using an SI of similar form adapted to gridded data. Heneka and Ruck (2008) presented an SI using the square of exceedances, although this assumed the damage threshold was normally distributed. Bloomfield et al. (2023) introduced a flood severity index, also using the exceedance over threshold approach, using linear exceedances of river flow data. SIs of this form are less influenced by outlier extreme events.*

Section 2.2 has had L96-L103 added:

*Aggregated Severity Indices (ASIs) are frequently used as a proxy for total damage. Hillier and Dixon (2020) aggregated wind gust and total precipitation over the extended winter season (October-March), concluding that extreme precipitation winters results in an uplift of aggregate extreme wind hazard for most of Europe. This compared the relative value of wind ASIs for the top and bottom thirds of winters ranked by precipitation ASI. Hunter et al. (2015) used ASIs to investigate the relationship between frequency and mean intensity of windstorms, concluding the Scandinavian Pattern was a driver of this relationship. Jones (2022) derived a framework to model the relationship between frequency and wind ASI, while Jones et al. (2024) found most of Europe have negative Pearson's correlation between wind and precipitation ASIs at high thresholds.*

*To increase the readability of the work and make it accessible to a wider audience, I think that an additional Figure panel is needed to illustrate the data underlying the negative correlation i.e. a scatter plot showing the ASI data points for a particular (x,y) location, the two thresholds from Fig2 a-c, different colours for dots contributing to Fig. 2a-c at the location. Consider adding a trend line for the correlation(s). And, also mark this location on Fig2 a-c. This might be a second panel in Fig. 4.*

Figure 4 had 3 panels added, showing the ASI for all of France, the correlation and the framework estimate for the 3 thresholds shown in Figure 2 (new figure shown in Figure A) Figure 4 has been amended to ensure calculation of framework components reflects the methods shown in Appendix A.

We have added "Jones et al. (2024) describes how sample correlation between wind and precipitation ASIs decreases with increasing threshold, including differing behaviours between regions.". (L195)

[Figure]

*Figure A - Panel (a) shows the framework (red line) with 95% confidence interval (red shaded area). Framework C is shown by the blue dashed line. Framework components shown by the thin black lines. Thresholds considered in Figure 2 shown by vertical lines. Panels b,c,d show scatter plot of ASI values at (0m/s,0mm), (10m/s,10mm and (20m/s,20mm) respectively.*

*L7 – Here, please make it clear that the assumption is of the same number of events for each hazard. Why might this be appropriate for extreme wind, precipitation and flooding - particularly when they exceed a threshold? This can be discussed later. The number of impactful flooding and wind events expected in the UK are, for instance, expected to differ (e.g. see Hillier et al 2024 - doi:10.5194/gc-7-195-2024). If I have mis-understood, please use this opportunity to make the paper more widely accessible i.e. is this handled by the distribution shapes of X and Y.*

For simplicity, our modelling approach uses only one count variable – the number of events where either wind or precipitation (or both) are extreme. This avoids the more complex problem of modelling the bivariate distribution of separate counts of when the wind and the precipitation are each extreme. The existing explanation (L89-92) in Section 2.2 has been amended to (now L108-112):

*In this study, we shall consider events that have perils caused by two hazard variables X and Y with thresholds $u_X$ and $u_Y$, respectively, resulting in annual ASI $S_X$ and $S_Y$. The total number of events, N, only includes events that increase $S_X$ or $S_Y$ (or both) i.e. events where $X > u_X$ or $Y > u_Y$. This avoids having to model the bivariate distribution of separate counts for extremes in wind and precipitation. The count variable used in this study is an upper bound for these separate counts. For simplicity of notation, we shall refer to $X' = g(X)$ and $Y' = g(Y)$ simply as X and Y, respectively.*

*L14 – Please fix the grammar here. Perhaps, '... high thresholds'.*

Changed to 'high thresholds'. (L14)

Response to Anonymous Referee #2

*L57 – Is it important that the metrics used for 'correlation' in this paragraph are mentioned, as the difference between how these measures change in the same (or different ways) is an important to the interpretation of the models in this paper in the context of previous work. To illustrate that metric matters, it is perfectly possible for spearman's correlation to decrease as tail impact increases (i.e. see material at https://doi.org/10.5281/zenodo.12533412, Q11&12 on Task 2). Hillier et al (2025) – 10.1002/JOC.8763 – have also quantified co-occurrence and use Uplift.*

Good point. The metrics used to measure association in each of the studies cited in this paragraph are now clearly stated in L97-L103. See response to the first comment for the revised paragraph.

This study uses Pearson's correlation to compute sample correlation values and as a basis to derive the frameworks. We acknowledge that Pearson's correlation is sensitive to outliers and it does not always capture the full picture of dependence. We have added L324-L325 to highlight this:

> *Pearson's correlation is used as a measure of dependency, this measure can be influenced by outlier events. Furthermore, zero correlation does not always imply independence (Embrechts et al., 2002)*

*L57 – Another detail that needs to be noted to allow the reader to compare this work to previous work is how annual/seasonal are defined i.e. Oct-Mar or similar in previous work, and Jan-Dec here, effectively taking part of two storm seasons.*

> Adapted so the correlation metrics for each study are mentioned and differences in timescales are highlighted (L64-65). Correlations between the ASIs for the extended winter are now shown in the appendix. Correlations computed over winter are more positive.

*L70 – Applying a correlation between hazard variables in order to understand potential losses is not, as far as I am aware common. It is done in one context by Hillier (2025). Please add references to put this choice of approach in context, particularly if they've been missed by Hillier (2025).*

> Line 70 (of the original manuscript) refers to the $\theta$ parameter of the framework. This correlation (between hazard variables) is a component of the overall framework which aims to capture the sample correlation ASIs. As far as we are aware, using constructing a framework for correlation between ASIs which uses the correlation between the hazards themselves is a new approach.

> As far as we can tell, Hillier et al. (2025) does not replicate this approach. Hillier et al. (2025) describes the ratio between two the number of events ("uplift") as correlation, comparing shorter and longer event windows.

*L77 – The phrase '.. following Jones et al (2024) ….' implies that Jones is the originator of this approach of SI over a threshold. Please add references and rephrase to better reflect the origin of the approach.*

Response to Anonymous Referee #2

*L79 – Similarly, the 20 ms-1 threshold is widely used and sensitivity tested (e.g. in Hillier & Dixon, 2020) although it long predates this. Please add references and rephrase to better reflect the origin of the approach.*

Removed "following Jones et al (2024)" and added "Klawa and Ulbrich (2003) were one of the first to use this threshold approach, noting German insurers usually pay for damages if a nearby weather station records gusts above 20ms$^{-1}$." on L89-90.

*L83 – Similarly, for Section 2.2 concisely add some background for the use of aggregate SIs for wind and cyclones.*

We have expanded these sections to add more background (L96-103).

*L105 – Fig. 1 is very helpful to improve accessibility. Thank you.*

Thank you - we are pleased to see that this has been helpful.

*L115 – Please clarify in words whether Z modulates X and Y in the same way (i.e. up or down), or whether it is possible for Z to affect X and Y differently.*

We have now added "and can influence X and Y differently" to L150.

*L160 – Following, or replicated from? Please clarify to make it clear whether Figure 2 is new work.*

Removed "Following Jones et al. (2024)". The sentence now reads "Figures 2a)-c) reproduce the sample correlation values at each grid point between wind and precipitation ASIs for different threshold levels shown in Jones et al. (2024)". (L200)

*L163 – Please add a sentence here on whether or not the spatial pattern of high and low correlations match previous work (e.g. Martius, 2016; Hillier & Dixon, 2020). If there is a match, it will then be clear to the reader that there is agreement spatially, with the primary debate about the magnitude & sign of the correlation.*

Added *"These findings differ to existing wind-precipitation research. This study uses aggregate scores while Martius et al (2016) and Owen et al. (2021) didn't, these are computed over the calendar year rather than seasons like Hillier and Dixon (2020). This study also links wind and precipitation to tracked cyclones while Martius et al., (2016) and Hillier and Dixon (2020) use daily data."* (L208-211).

*L193 – Red box for France. Is this 5° from the central point? If not, it would be better for comparison and consistency across the manuscript to make this match the approach on L157, and clarify there that it's a box not a radius.*

The France box refers to the region of interest. Any storm that comes within 5° of any land gridpoint within the box is considered. However multiple gridpoints within the box will have a SI from one storm event. To calculate one SI value for the France box, the SI for all of the gridpoints over land are summed. This is done for wind and then again for precipitation, giving each hazard one SI value for the entire region. This is now more carefully explained in the text.

Changed to read: *"For each storm, the SI for the region is calculated by summing the SI from all land and sea grid points in [4.75°W-8.5°E, 42.25°-51.75°N]. For a given storm*

*most gridpoint SIs within the region are zero, being >5° from the storm track or below the threshold."* (L241-243)

*L206 – This jet stream connection was first published in Hillier & Dixon (2020) - Dixon's idea. Please consider add the reference to this point.*

Changed to: *Hillier and Dixon (2020) first proposed that a weaker jet stream is conducive to precipitation-only extremes. Manning et al. (2024) also concluded slow moving windstorms and a weaker jet stream are favourable for precipitation-only extremes. (L260-263)*

*L235 – Fix brackets for references.*

Fixed, thank you for spotting this error. (L303)

*L251 – To link this work into that in the introduction, it is probably worth showing the Oct-Mar results in Supplementary Material so that readers can decide how to link this to earlier work.*

Figure B has been added to the supplementary material, this shows framework estimate (a-c) for the sample correlation (d-f) for all storms with genesis times in the extended winter. This splits years from October-March by cyclone genesis time.

[Figure]

*Figure B - Framework C estimate (a-c) of sample correlation (d-f) for ASIs calculated over the extended winter (1st October-31st March). Sample correlation not significant at the 5% level is shown by stippling.*

*Figure 2 – On a-c, please add stippling or similar to indicate areas where the observations have statistical significance (i.e. can be safely interpreted as existing). And, in the caption, decide if units are in italics or not.*

This would add too much noise and reduce the message of the plot, however we have added a plot showing the framework and sample correlation with stippling to the supplementary material. The units in the captions are now in italics.

*Figure 4 – Add error bars, or similar, to the sample correlation curve please.*

A 95% confidence interval for sample correlation has been added.

*Figure 5 – The two sets of numbers on the scale bar reduce clarity. Please find another solution. However, I like this figure; interesting similarities to Fig. 2g of Hillier & Dixon (2020) showing different wind directions for 'windy' and 'wet'.*

The black bars showing days have been removed. The similarities to Fig. 2g of Hillier & Dixon (2020) are now mentioned in the text (L275-L277):

*Hillier and Dixon (2020) found a similar contrast when considering wind direction on windy and wet locations days. Wind directions at a site on Scotland's east coast were south-westerly on days with extreme wind but north-easterly on days with extreme rain.*

*Figure 6 – Please add stippling or similar to indicate where the correlations are statistically significant (or not)*

This would add too much noise and reduce the message of the plot, however we have added plots with stippling to the supplementary material.

---

## Author Comment (AC2)

Response to Anonymous Referee #3

We would like to thank the reviewer for their helpful comments on our manuscript. Below we list how we've addressed each of their comments (in blue italics).

*(Methodology) I think this section would benefit with some clarification and simplification. The description of the approach is quite mathematical and may be difficult for some readers of the journal to follow. Given the potential value of this method as a tool for the wider community, simplifying certain explanations could encourage broader uptake. These are mostly suggestions, though there are places where I feel I could not replicate without making assumptions. I leave it to the authors to decide whether these changes would strengthen the text.*

> Thank you for this helpful comment. We have now added more descriptive detail around the assumptions (L128-133), this reads:
>
> *The distinction between the frameworks are their choice of independence assumptions:*
>
> *– Frequency-Severity Independence (FS-Ind): The severity of hazards within a year are independent to the frequency of events (e.g. storm counts and gust speeds for a year have zero correlation).*
>
> *– Hazard Independence (H-Ind): The wind and precipitation SIs from the same event are assumed independent (so have zero correlation).*
>
> *– Serial Independence (S-Ind): Hazard SIs from different events are assumed independent (wind values from separate events have zero correlation with each other).*

*P5 L108-111, the subscript j is introduced without explanation. Furthermore, why use Cov() when var(X) is used later on L126? Although they are interchangeable, using only one would be consistent and enhance readability.*

> We have acknowledged the *j* index on L138:
>
> *As indices i and j take any value from 1 to N , dependency between all hazard pairs within a year is considered.*
>
> By definition, variance and covariance are not interchangeable. However the variance is a special case of covariance, (Cov(X,X)=Var(X)). For ease of interpretation we will continue to use both variance and covariance.

*P5, L113-114: For the correlation , should this not be ? Likewise for P5, L131. I perceive  as the correlation of one pair of X and Y values which is obviously not the case. I assume that the correlation is calculated between all X and Y values? Furthermore, it would be helpful to to use , to indicate individual events where appropriate, and X and Y to indicate the random variables.*

> Correlation ($\theta$) is calculated for all X and Y values, which is now clarified on L144-145.

*P5, L114: Which measure of correlation is used?*

> Updated to clarify that Pearson product moment correlation was used, L144.

Response to Anonymous Referee #3

*P5, L125: Please clarify how  and  are calculated. I assume it is via the same equations on P6, L138 without conditioning on Z.*

The original manuscript defined $J_X$ and $J_Y$ on L126 (now L159). The appendix has been updated to include the E[X] and Var(X) paramter estimates which do not condition on $Z$ (L353-363).

*P6, L136-139: I suggest providing some plain explanation on what these terms represent.*

In the original manuscript this is provided later on L180-184 (in 3.3 Analysis of components in Framework C). It is now L223-228. Explanations were not given at this this point in the manuscript as it does not make sense to consider framework components when we do not know how the framework performs. We have signposted that these terms are explained later (L173):

*Physical explanation of these three components is provided in Section 3.3.*

*Minor comments:*

*P6 L146-148: Could you clarify what you mean by 'smaller scale features'? ERA5 is unlikely to reproduce certain features such as sting jets and does not resolve convective rainfall, which are the features I would consider smaller scale. Also, I would double check the cited papers here as I don't see a mention of ERA5 within them, and one paper predates ERA5.*

Yes, sting jets are not  resolved in ERA5, this has been removed. This sentence was highlighting the use of high resolution hourly data will achieve better results than use of daily data. It has been reworded to say this and not imply that the references use ERA5 data when they don't (e.g. Whitford et al. (2023)).  The sentences (L180-182) now read:

*"The cyclones and hazard variables are extracted from 1-hourly ERA5 reanalysis at the native 0.25° spatial resolution from1980-2020 (Hersbach et al., 2020). Using hourly data is important for modelling sub-daily rainfall extremes (Whitford et al., 2023)."*

*P6 L156: Do you mean the maximum 3-second wind gust instead of the '3-second maximum'?*

Yes, this has now been changed (L190)

*P6 L160: Please clarify which measure of correlation is used.*

Clarified Pearson's product moment correlation is used. (L196)

*P6 L163: Are the thresholds applied to the event metrics and not the hourly?*

Yes, this is now clarified in Section 3.1 (L192).

*Figure 2: Could you include the motivation for using absolute thresholds instead of percentile thresholds? Also, you mention in the conclusions that the results are insensitive to this choice (P13 L246-247), but it would be informative to provide a map in supplementary showing what*

*percentile the absolute thresholds here fall under. I can't tell if 10mm or 20mm in a storm is extreme or not.*

Fixed thresholds were used as this paper builds on results of Jones et al. (2024) where fixed thresholds are used. We have added a figure to the Appendix showing the percentiles that (10ms$^{-1}$,10mm) and (20ms$^{-1}$,20mm) fall under (this figure is shown below). This motivation is now explained on L197.

[Figure]

*Figure 1: Equivalent percentiles for wind gusts (a & b) and precipitation (c & d) for thresholds u=10 (a \& c) and u=20 (b & d).*

*P8 L176: Framework C Indicates the presence of a distinct land-sea contrast in the correlations for joint exceedance of 20 ms-1 and 20mm, while there are indications of this in the sample correlation. Can you comment on why we see the land-sea contrast and why the sample correlations are noisier?*

The land-sea contrast is primarily caused by the fact that wind speeds are generally greater over sea (reduced surface roughness) and so wind exceedances over a fixed threshold over sea are less in the extreme tail of the distribution than those over land. The negative correlation emerges over sea at higher thresholds (e.g. above 40m/s for the Azores in Fig. 6 of Jones et al. 2024). This is now mentioned in Section 3.2 (L204-206). We suspect that the sample correlations look noisier because their values are slightly closer to zero over land.

*P8 L180: Could you clarify what is meant by 'simultaneous correlation'? Do you mean the local correlation each grid cell?*

This has been removed and redefined as: "the 'average' of yearly wind-precipitation correlation, computed between hazard pairs occurring from the same storm." (L227)

*P8 L187-188: Should 3g and 3i be 3d and 3f?*

Yes, this has now been changed. (L236)

*P8 L197-198: I think there is a typo here:"The positive within year dependency component (solid thin line) is largely compensated at all temperature thresholds by the negative within year dependency component", i.e. "within year dependency component" is repeated twice.*

Yes, thank you. This has been changed to "*The positive event dispersion component (solid thin line)...*" (L247)

*Figure 4 and 5:*

*The negative within-year correlations (dashed line) appear much stronger over this area of France than the grid-cell correlations shown in Figure 3d–f, which are close to zero across the domain. Could you comment on why this might be the case? Does the larger domain introduce spatial correlations between wind and rainfall? This is an interesting result and may relate to the findings of Manning et al. (2024), already cited here, given the cancelling effect on the dispersion component. For example, one might expect a negative spatial correlation on an event basis, as the highest rainfall typically occurs to the northeast of a cyclone centre, while the strongest winds occur to the south. This spatial separation could explain the cancelling influence on N.*

As SIs are aggregated over a larger domain there is greater spatial distribution of hazards for the France region. For the 98th percentile, storms are typically windier in the north west and wetter in the south east (shown in Figure 2c & e). At higher thresholds the SI for the region tends to be extreme for just one hazard, giving negative correlation.

Figure 3 shows splitting the France region into north and south regions. Figures 4 and 5 are the same as Figure 4 in the manuscript but recalculated on these regions. As expected the value of the within-year correlations is lower when splitting the regions, but more negative for the northern region.

*You see a similar effect of the relative positioning of wind and rainfall extremes in cyclones in Figures 5b and 5c. Most wind-event tracks pass to the north of the domain, exposing a larger portion of the domain to the part of the cyclone that generates strong winds. Conversely, rainfall-extreme tracks tend to lie farther south, meaning a greater part of the domain overlaps with the cyclone sector producing heavy rainfall. See Figure 3 in Manning et al. (2024) and related discussion that shows similar results.*

*Overall, I think the manuscript would benefit from further discussion linking the statistical results to the underlying physical processes, as well as examining the sensitivity of the findings to domain size (e.g., grid cells in Figure 3 versus the larger domains in Figures 4 and 5). The paper presents interesting results on the propagation speed of systems, but I believe there are further nuances to discuss, such as storm positioning, as noted above. It would also be valuable to extend the discussion to consider the influence of the jet stream, which is a dominant driver of propagation speed.*

The manuscript has been updated to comment on the sensitivity of results relative to domain size (L251-254). We have expanded the discussion to highlight similar results found in Manning et al. (2024) and Owen (2022) (L281-286). We feel discussion regarding the jet stream is beyond the scope of this study, although have highlighted Hillier et al. (2025) on L278.

*Title: I suggest amending the title to highlight its application to wind and rainfall extremes. While the paper presents an excellent tool, it also makes a valuable contribution to the understanding of multivariate wind and rainfall extremes which might otherwise be overlooked.*

The title has been updated to "Collective risk modelling of multi-peril severities: the correlation between European windstorm annual accumulated gust speed and precipitation exceedances".

[Figure]

*Figure 2 – Percentile values for wind gusts (a-c) and precipitation (d-f) for 0th (a&d), 80th (b &e) and 98th (c & f) percentile thresholds.*

[Figure]

*Figure 3 - North (blue) and South (red) France regions used in Figures 3 & 4.*

[Figure]

*Figure 4 - Same as the article's Figure 4 but computed for just the Northern half of the France region (shown in Figure 2 above)*

[Figure]

*Figure 5 - Same as the article's Figure 4 but computed for just the Southern half of the France region (shown in Figure 2 above)*

---

## Author Comment (AC3)

Response to Anonymous Referee #1

We would like to thank the reviewer for their helpful comments on our manuscript. Below we list how we've addressed each of their comments (in blue italics).

*I was intrigued by the title of the paper which states that it investigates multi-peril accumulated losses. Therefore, I was surprised to find that there is in fact, no loss data used in the paper. The paper only looks at the 'severity' purely based on ERA5 data of wind and precipitation. 'Losses' may cause the reader to expect a more quantifiable impact, such as damages, or have some sort of vulnerability or exposure included in the loss function. I understand that the authors is using severity as a proxy for loss, however, the hazard already seems to be a proxy of the severity through the use of a function. Perhaps the authors can consider to change the word losses to severity instead, in order to avoid further confusion.*

This is a good point to make and so we have replaced the word "losses" to "severity" in the article title. The title is now:

*"Collective risk modelling of multi-peril events: correlation of European windstorm gust and precipitation annual severity"*

*In Line 173, the authors mention that "negative correlation over the northwest of mainland Europe is correctly captured" and only framework C is able to capture the correlation at each threshold. It seems that it is 'correct' because it matches the sample correlation? What makes the sample correlation from Jones et al. 2024 'correct'?*

We are aiming for the framework to capture the broad spatial features in the sample correlation (shown in Jones et al. (2024)) rather than reproduce exactly the grid point values which are prone to sampling uncertainty. Our use of the word "correctly" is therefore confusing. L173 (now L216) has now been changed to read:

*"The spatial structure of negative correlation over the northwest of mainland Europe is broadly reproduced at the highest thresholds (Figure 2k)"*

*I wonder why only data from 1980-2000 has been investigated, while the data is available to the present. The authors state that data prior to 1980 has not been included due to data quality, however, this should not be an issue for recent data. Does the cutoff in 2000 mean that the current day climate is not reflected in the results?*

Sorry – this was a typographical error in the abstract that has now been corrected to "1980-2020".

*The introduction of the paper reads very well and highlights the general need for the proposed research. I find the findings related to the negative correlations and the difference between storms with a short and long duration interesting. However, as a reader I am left wondering exactly why these findings are important. Who is it relevant for? How may these results improve our multi-peril risk management? Additionally, a negative correlation between wind and rain can already be deducted by looking at Figure 5, which provides enough insights for the relationship between hazard intensity and duration as well. Can the authors explain the need of extensively testing the three frameworks that are presented as the main focus of the paper?*

It is relevant to risk managers such as catastrophe reinsurers who could in principle exploit the negative correlation (if it exists at a portfolio level between wind and flood losses) to help diversify their risk. For example, they could try to balance windstorm cover with flood insurance cover in countries such as France.

Figure 5a only shows SI for individual events and this, as framework C shows, is not necessarily sufficient to guarantee negative correlation in annual "aggregate" SI across the year (positive correlation from clustering could cancel it). The frameworks are important for understanding the causes of the resulting correlation in ASI.

These explanations have now been added to Section 1 (L55) and Section 3.2 (L198) of the manuscript.

---

## Referee Report (RR1)

Thank you for engaging fully with my comments and providing good justifications for why you have or have not made changes. I think that the changes made improve the manuscript significantly, making it easier to understand and more transparent to the NHESS readership. I believe that this paper is now suitable for publication, subject to two minor changes noted below. Admittedly, one remains from the original comments due to my provision of inaccurate information on the reference date.

- L70 in the original manuscript, L74 in the revised manuscript. Apologies for the inconvenience, I intended to refer to Hillier (2024) [https://doi.org/10.5194/gc-7-195-2024] instead of Hillier (2025) with my comment about previous uses of statistics to link annually aggregated measures of hazard. Please see the figure below. Admittedly, the use of copulas to link ranks of hazards is significantly simpler than the statistics presented by Jones et al, which leaves a lot of space for novelty. Indeed, calling Hillier (2025) a 'framework' might be over-stating it, but it did establish the idea of linking hazard variables to consider the impact on losses, even if only in one narrow context. Since the authors say they know of no other work missed by Hillier (2025) adding a sentence (or even half a sentence) could readily account for this.

[Figure]

- Thank you for adding stippling to figures showing statistical significance in the supplementary material. Please add a cross reference to the captions of Figure 2 and Figure 6, directly pointing the readers to the new figures in the supplementary material that indicate the regions where observations are statistically significant.

To allow others to follow their work, the authors might consider making illustrative code available (e.g. to process annual data from a single x,y point), but I believe this should be regarded as optional.